# Plasma-Induced Changes in the Metabolome Following Vistula Tart Cherry Consumption

**DOI:** 10.3390/nu16071023

**Published:** 2024-04-01

**Authors:** Emma Squires, Ian H. Walshe, William Cheung, Samantha L. Bowerbank, John R. Dean, Jacob Wood, Malachy P. McHugh, Stephan Plattner, Glyn Howatson

**Affiliations:** 1Faculty of Health and Life Sciences, Northumbria University, Newcastle upon Tyne NE1 8ST, UK; e.squires@northumbria.ac.uk (E.S.); ian2.walshe@northumbria.ac.uk (I.H.W.); william.cheung@northumbria.ac.uk (W.C.); samantha.bowerbank@northumbria.ac.uk (S.L.B.); john.dean@northumbria.ac.uk (J.R.D.); jacob2.wood@northumbria.ac.uk (J.W.); mchugh@nismat.org (M.P.M.); 2Nicholas Institute of Sports Medicine and Athletic Trauma, Northwell Health, New York, NY 10065, USA; 3Iprona Lana SpA, 39011 Lana, South Tyrol, Italy; 4Water Research Group, North West University, Potchefstroom 2531, South Africa

**Keywords:** tart cherry, metabolome, bioavailability

## Abstract

Evidence suggests that tart cherry (TC) supplementation has beneficial effects on health indices and recovery following strenuous exercise. However, little is known about the mechanisms and how TC might modulate the human metabolome. The aim of this study was to evaluate the influence of an acute high- and low-dose of Vistula TC supplementation on the metabolomic profile in humans. In a randomised, double-blind, placebo controlled, cross-over design, 12 healthy participants (nine male and three female; mean ± SD age, stature, and mass were 29 ± 7 years old, 1.75 ± 0.1 m, and 77.3 ± 10.5 kg, respectively) visited the laboratory on three separate occasions (high dose; HI, low dose; LO, or placebo), separated by at least seven days. After an overnight fast, a baseline venous blood sample was taken, followed by consumption of a standardised breakfast and dose conditions (HI, LO, or placebo). Subsequent blood draws were taken 1, 2, 3, 5, and 8 h post consumption. Following sample preparation, an untargeted metabolomics approach was adopted, and the extracts analysed by LCMS/MS. When all time points were collated, a principal component analysis showed a significant difference between the conditions (*p* < 0.05), such that the placebo trial had homogeneity, and HI showed greater heterogeneity. In a sub-group analysis, cyanidine-3-*O*-glucoside (C3G), cyanidine-3-*O*-rutinoside (C3R), and vanillic acid (VA) were detected in plasma and showed significant differences (*p* < 0.05) following acute consumption of Vistula TC, compared to the placebo group. These results provide evidence that phenolics are bioavailable in plasma and induce shifts in the metabolome following acute Vistula TC consumption. These data could be used to inform future intervention studies where changes in physiological outcomes could be influenced by metabolomic shifts following acute supplementation.

## 1. Introduction

Polyphenols are well-known bioactive compounds, widely occurring in plants, and are noticeable for their deep purple/red colouring. Polyphenol-rich foods have attracted attention for having both antioxidant and pro-oxidant activities in vitro and in animal models [1]. There are many reported health-promoting effects from consumption of polyphenol-rich foods, such as anti-inflammatory, antioxidant, antiestrogenic, cardioprotective, chemoprotective, and neuroprotective properties [2,3,4]. Previous studies have investigated different polyphenol-rich foods and how these compounds might impact health. These include decreased risk of all-cause and cardiovascular disease mortality [5,6], improved sleep and reduced stress [7], and being inversely associated with hypertension [8,9].

Several studies have demonstrated tart cherries (TC) to have a wide range of benefits, spanning across health paradigms that included improved cardiovascular risk factors [10], mood, urinary cortisol, stress and anxiety [11], as well as sleep quality and duration [12], and reduced blood pressure [13,14]. In exercise models, TC have reduced exercise-induced pain [15,16] and exercise-induced inflammation [17] and led to greater return of muscle function following strenuous exercise [18,19,20]. Furthermore, studies have suggested TC to be a source of bioavailable anthocyanins and polyphenols, such as flavonoids, kaempferol, quercetin, catechin, and melatonin [12,21,22,23,24] and therefore provide support that these compounds positively affect human physiology.

Polyphenols are poorly absorbed in the small intestine. Colonic microbiota (the collection of microbes living in the large intestine) break them down into smaller metabolites accessible to the host [25,26,27], which has been demonstrated in both human and animal studies [23,28,29]. Keane, Bell, Lodge, Constantinou, Jenkinson, Bass, and Howatson [23] reported plasma concentrations of phenolic acids to be most abundant 1–2 h post initial ingestion with traces detectable at 8 h post ingestion, following TC consumption. Similarly, Kirakosyan, Seymour, Wolforth, McNish, Kaufman, and Bolling [29] observed high concentrations of anthocyanins in rodent liver, bladder, kidney, and brain tissue following TC consumption. Both studies provided evidence to suggest polyphenols and anthocyanins are metabolised quickly, similar to phenolic acids, or have the potential for bioaccumulation in some tissues.

When attempting to determine the mechanisms by which these effects occur, the research is limited. However, research in the human metabolome [30,31] found TC consumption improved the ability to sustain attention, due to increased amino acid circulation which has been associated with cognitive function. These data have focussed almost exclusively on the Montmorency cultivar of TC; there are no data on the more widely abundant Vistula TC cultivar, despite being reported to have high concentrations of polyphenols and anthocyanins [32]. Since it is similar to the Montmorency variety, it makes the expectation tenable that consumption of Vistula TC would also cause observable changes in the metabolome. Therefore, the aim of this study was to examine the effects of acute Vistula TC consumption on changes in the human metabolome.

## 2. Materials and Methods

### 2.1. Participants

Twelve healthy participants were recruited for this study: nine males, where mean ± SD age, stature, and mass were 31 ± 7 years old, 1.76 ± 0.1 m, and 78.7 ± 8.2 kg, and three females, who were 23 ± 3 years old, 1.72 ± 0.14 m, and 73.2 ± 15.4 kg, respectively. All participants were healthy, as assessed by a health-screening questionnaire and provided written, informed consent. Exclusion criteria for the study were as follows: food allergy (as discussed with the research team), history of gastrointestinal, renal, cardiovascular, or thyroid disease and current use of any food supplementations. All subjects gave their informed consent for inclusion before they participated in the study. The study was conducted in accordance with the Declaration of Helsinki, and the protocol was approved by the Ethics Committee of Northumbria University Ethics Committee (reference number: 39904 and date of approval 5 January 2022), and it was registered as a clinical trial with clinicaltrials.gov (NCT06144489).

### 2.2. Experimental Design

The study utilised a double-blind, three-arm, cross-over, pseudo-randomised (counter-balanced to eliminate order effects) design to identify metabolomic shifts following the acute ingestion of a LO and HI dose of Vistula TC and placebo. As a first step, the content of the supplement was analysed. A sub-group of the participants’ (*n* = 6) data were analysed for bioavailability of potential polyphenols. This was followed by examining the metabolomic profile in plasma. A washout period of at least 7 days (but not greater than 21 days) between each phase was implemented. Participants were required to attend the start of each phase of the study at 7:45 a.m. following a 10 h overnight fast to account for diurnal variation. Upon arrival at the laboratory, the participants’ baseline blood pressure (BP) was recorded in a rested state, then cannulated for serial blood sampling. A standardized breakfast was then provided. Subsequent measures were then taken at baseline, 1, 2, 3, 5, and 8 h post Vistula TC consumption. Water was provided ad libitum, but no additional food was given during testing visits.

### 2.3. Treatments and Dietary Control

The LO and HI dose consisted of 3 or 6 g, respectively, of the TC spray-dried extract made from a variety of TC known as ‘Nadwiślanka’, also called Vistula Cherries (extract brand name CherryCraft^®^, Iprona Lana SpA, Lana, Italy). The placebo given was a 3 g capsule matched for caloric content using fructose, glucose, and maltodextrin. The doses provided were in line with previous work on bioavailability with a TC concentrate [23]. The LO TC dose contained approximately 60–78 mg of anthocyanins (8.07 kcal and had a macronutrient content of 1.215 g, 0.003 g, and 0.042 g for carbohydrates, fat, and protein, respectively), and the HI TC dose approximately 120–156 mg of anthocyanins (16.14 kcal with 2.43 g, 0.006 g, and 0.080 g for carbohydrates, fat, and protein, respectively), which is comparable to other work in this area [19,23,33,34,35].

To increase external validity, participants were requested to follow their habitual diet throughout the trial. Food diaries were completed for the 24 h before each testing visit to replicate (as close to) the conditions prior to testing on following testing visits. The standardised breakfast consisted of 2 slices of toasted white bread and ~16 g of a commercially available spread. Participants were given water to drink ad libitum throughout the testing visits, this was measured and refilled from baseline, and 1, 2, 3, 5, and 8 h post breakfast and supplement ingestion.

### 2.4. Blood Pressure Monitoring and Blood Sampling

Blood pressure was monitored throughout the testing visits, immediately upon arrival to the laboratory and 1, 2, 3, 5, and 8 h following ingestion of the treatment (LO, HI, or placebo) [14]. Venous blood samples (~10 mL) were collected from the antecubital fossa into lithium heparin (10 mL) tubes, immediately upon arrival to the laboratory and 1, 2, 3, 5, and 8 h following ingestion of the treatment. Samples were immediately centrifuged (3000× *g*) at 4 °C for 20 min; plasma was then aspirated and pipetted into ~1 mL aliquots and then immediately stored at −80 °C for later analysis.

### 2.5. HPLC Analysis

Plasma samples were thawed prior to extraction. Aliquots of 500 µL were decanted to microcentrifuge tubes along with 500 µL of 10% formic acid in methanol. Samples were vortexed and centrifuged at 5300× *g* for 2 min. The supernatants were loaded onto a HyperSep™ C18 solid phase extraction (SPE) cartridge (Thermo Scientific, Hemel Hempstead, UK), which were preconditioned sequentially with 1 mL of 10% formic acid in methanol and 1 mL 10% formic acid in water. The cartridge was eluted with 500 µL of 50:50, *v*/*v* methanol in water with 10% formic acid. The eluent was dried down under a stream of nitrogen and reconstituted in 100 µL of 10% formic acid in methanol.

HPLC analysis on the plasma samples, for quantitative analysis of 3,4-dihydroxybenzoic acid, chlorogenic acid, VA, C3G, C3R, peonidin-3-*O*-rutinoside and melatonin, was performed using an Agilent Infinity HPLC (Agilent Technologies, Cheadle, UK) consisting of a quaternary pump, an autosampler, a thermostated column oven (set to 40 °C), and a multiwavelength detector monitoring at 247 nm. The system was operated using OpenLab chromatography data system (Agilent Technologies, Cheadle, UK). Chromatographic separation was achieved on a reversed phase Eclipse Plus C18 column (3.5 µm, 100 × 4.6 mm) from Agilent (Stockport, UK). Sample aliquots of 10 µL were introduced onto the column at a flow rate of 200 µL/min. The analytes were separated using 87:3:10, *v*/*v*/*v*, water/acetonitrile/formic acid (%*v*:%*v*:%*v*) (A) and 40:50:10, *v*/*v*/*v*, water/acetonitrile/formic acid (B) as the mobile phase using a gradient (Appendix A).

### 2.6. Untargeted Plasma Metabolomic Analysis

The samples analysis was performed at Northumbria Metabolomic Core Service, Northumbria University. The extraction and acquisition parameters were as follows.

#### 2.6.1. Sample Extraction

The sample was prepared by firstly being thawed on ice; then, 200 µL of plasma was extracted and vortexed with 1000 µL of analytical grade methanol, then sonicated in a water ice bath for 15 min. The samples were then centrifuged for 15 min at 15,000 rpm at 4 °C, before the supernatant was collected and dried down in a vacuum pre-concentrator for 2 h at 45 °C. The plasma extracts were reconstituted in 100 µL of LC/MS grade water (initially vortexed for 30 s and sonicated for 15 min at 4 °C) before being filtered via 0.22 micron Costar spin filter and put in a 1.5 mL autosampler vial with a 200 µL microinsert.

A total of 20% of all samples were pooled together to create quality controls for analytical stability assessments.

#### 2.6.2. Data Acquisition

Metabolite characterisations were performed on a Thermoscientific LC/MS system (Vanquish Liquid chromatography Front end connected to IDX High Resolution Mass Spectrometer system, Thermo Scientific, Hemel Hempstead, UK). The MS data were acquired using Thermo’s AcquieX acquisition data dependent analysis methodology. The MS operating parameters were as follows: MS1 mass resolution 60K, for MS2 30K stepped energy (HCD) 20, 25, 50. *m*/*z* scan range 100–1000, RF len (%) 35, AGC gain, intensity threshold 2 × 10^4^ 25% custom injection mode with a maximum injection time of 50 ms. An extraction blank was generated and used to create a background exclusion list, and pooled quality control (QC) was used to create the inclusion list (peaktables are available in the Appendix A). Chromatographic separation was achieved using a Waters Acquity UPLC T3 HSS column (Waters Limited, Wilmslow, UK; 2.1 × 150 mm × 1.7 μm), operating at 45 °C with a flow rate of 200 μL/min. The LC gradient consisted of a binary buffer system, buffer A (LC/MS grade water/acetonitrile, 95:5, *v*/*v*) and buffer B (LC/MS grade acetonitrile/water, 95:5, *v*/*v*). An independent buffer system was used for positive and negative ion mode acquisition; the pH of buffers was adjusted using 0.1% formic acid for both. The LC gradient was the same for both polarities, 5% B at T_0_ hold for 1.5 min and linearly decreased to 95% B at 11 min, hold for 4 min and then returned to the starting condition with a hold for a further 4.5 min (column stabilization). The voltage applied for positive mode was 3.5 kV, and the injection volume was 3 μL. The HESI condition was as follows for 200 μL: sheath gas 35, aux gas 7, and sweep gas of 0. The ion transfer tube temperature was 300 °C, and the vaporizer temperature was 275 °C. Available standards for C3G, C3R, peonidin 3-glucoside were purchased from Biosynth (Staad, Switzerland), and chlorogenic acid, protocatechuic acid, VA, and melatonin were purchased from Sigma-Aldrich (St. Louis, MO, USA).

Due to the low concentration of melatonin present within the TC samples, a separate absolute quantification (AQ) methodology was applied using the same analytical methodology as described above. The AQ was performed using a 6-point calibration curve (5 µg/mL, 2.5 µg/mL, 1.25 µg/mL, 0.625 µg/mL, 0.3125 µg/mL, and 0.1562 µg/mL, respectively), *n* = 6 per concentration with R square value of 0.9905. One µL injection was used for each concentration. Twenty mg TC samples were extracted in 1 mL of analytical grade methanol; the subsequent extracts were processed using the same methodology as the plasma samples. Ten µL was used based on the calibration curve analysis (Appendix A).

### 2.7. Data Analysis

Statistical analysis was performed using SPSS (SPSS version 28, Inc., Chicago, IL, USA). Descriptive statistics are reported as mean ± SD. The BP was analysed using a treatment (LO vs. HI dose vs. placebo) by time (baseline, 1, 2, 3, 5, and 8 h post supplement) with repeated measurement of analysis of variance (RMANOVA). Mauchly’s test of sphericity was used to check homogeneity of variance for all variables; where necessary, any violations of the assumption were corrected using the Greenhouse–Geisser adjustment. Significant interaction effects were followed up using post hoc analysis. The alpha level for statistical significance was set at *p* = 0.05.

The LC/MS positive data set was processed via Compound Discoverer version 3.3 according to the following settings: Untargeted Metabolomic workflow: mass tolerance 10 ppm, maximum shift 0.3 min, alignment model adaptive curve, minimum intensity 1 × 10^6^, a S/N threshold of 3, compound consolidation, mass tolerance 10 ppm, and a RT tolerance of 0.3 min. Database matching was performed at MS2 level using Thermo Scientific *m*/*z* cloud with a similar index of 70% or better. In the positive mode, 83 metabolites were identified and quantified with an MS/MS match of 70% or greater. The QC RSD of the whole analysis was 7.62%.

## 3. Results

Firstly, there were no significant differences between the conditions or over time for systolic BP (*p* > 0.05) and diastolic BP (*p* > 0.05). Participants reported no side effects from all doses.

### 3.1. Analysis of the TC Extract

An example of the LCMS chromatogram of C3R, the standard and the supplement, is presented in Figure 1. The analysis of the extract revealed the presence of polyphenols (Table 1). AQ of melatonin found in the extract was 30.5 ng/mg.

### 3.2. Bioavailability Results

In a sub-group analysis (*n* = 6), the bioavailability of compounds identified in the extract was investigated in plasma (Figure 2). There were significant differences between all the conditions for both C3G and C3R (*p* < 0.05), as well as VA between the LO and placebo condition (*p* < 0.05). Due to sampling error, the HI dose for VA is not displayed. Area under the curve (AUC) for HI, LO, and placebo was 1.35, 0.91, and 0.61, for C3G, respectively, and 1.92, 1.27, and 0.47, for C3R, respectively. AUC for VA was 6.13 LO dose and 2.08 placebo.

### 3.3. Metabolomic Analysis

When all time points were collated, principal component analysis (PCA) showed changes (*p* < 0.05) between conditions; the placebo trial had greater homogeneity, such that participants showed similar profiles, whereas HI showed greater heterogeneity, where a marked shift in the metabolome was evident. The magnitude of this shift in the HI dose was varied and illustrates the variability of the response between volunteers. In addition, when conditions were examined at each time point, this trend was evident up to 8 h post supplementation (*p* < 0.05). The LO dose consistently overlapped both the HI and placebo dose throughout all time points. PCA analysis is displayed for all time points (Figure 3) post acute ingestion of a HI and LO dose of Vistula TC and placebo.

Partial least squares discriminant analysis (PLS-DA) provided the top-ranked discriminatory metabolites responsible for the changes in the metabolome; an example is provided for 1 h (Figure 4). Metabolites such as L-cystine, cortisone, and uracil are highly informative of the HI dose at 1 h (VIP score > 1.8). Class average heatmaps display the top 40 metabolites at 1 h (Figure 5). PLS-DA and class average heatmaps for all other timepoints are presented in Appendix A.

## 4. Discussion

This study aimed to examine the effects of acute Vistula TC consumption on changes in the human metabolome. This is the first evidence that Vistula TC (cultivar ‘Nadwiślanka’) contains phenolic compounds (Table 1), such as C3G and C3R, which are bioavailable in human plasma (Figure 2), and that a LO and HI dose of Vistula TC induces changes in the human metabolome over 8 h (Figure 3, Figure 4 and Figure 5).

Vistula TC contains phenolic compounds, VA, protocatechuic acid, chlorogenic acid, C3R, C3G, and melatonin (Table 1). Similar compounds have been identified in previous work [36,37] and concur with these current findings; both sour and sweet cherries have a high polyphenol content that is detectable. However, it was important to identify the unique composition of the Vistula variety, since there are differences between varieties that might underpin potential physiological changes. Kim, Heo, Kim, Yang, and Lee [37] found TC had cyanidin 3-glucosylrutinoside as the major anthocyanin, whereas sweet cherries had C3R as the major anthocyanin. Moreover, environmental factors such as precipitation, temperature, and sunlight will influence the levels of phytochemicals in TC [38]. Importantly, these data provide evidence that the European variety of TC provides a bioavailable source of phytochemicals (Table 1). This provides direct evidence that the phytochemicals ingested can be absorbed and detected in the blood suggesting it could be utilised in physiological processes.

These results showed that C3G, C3R, and VA were detected in plasma following acute consumption of Vistula TC, compared to the placebo group. This is in agreement with research by Keane, Bell, Lodge, Constantinou, Jenkinson, Bass, and Howatson [23], who also found phenolic acids (VA and chlorogenic acid) to be bioavailable, following consumption of Montmorency TC concentrate. Although caution should be taken with making direct comparisons because of the different methodologies employed, the amount of VA detected in the current study was greater than the aforementioned study (at 2 h post 1.191 µg/mL vs. >0.3 µg/mL). Another consideration is the type of TC product, as previous work used 60 mL of TC concentrate [23], containing a total anthocyanin content of 62.5 ± 0.3 mg/L, whereas the current study used a powdered extract containing 60–78 mg anthocyanin in the LO dose (3 g); although the anthocyanin content is similar, the powdered extract is a higher concentration. It has been previously demonstrated that powered Montmorency TC contains greater levels of anthocyanins in comparison to Montmorency TC concentrate [28]. This aligns with the present study’s findings which suggest the composition of Vistula TC may have greater bio-accessibility. There is typically a wide inter-subject variability, as phenolic compounds are historically difficult to detect in plasma, because they are both metabolised quickly and absorbed poorly [39]. Farah et al. [40] found a 7.8–72.1% uptake range in chlorogenic acid, which might explain differing results between studies. Another limitation is that analytic chemistry methodologies can vary between studies, so direct comparisons should be treated with caution. Despite that, these results show Vistula TC provides detectable and quantifiable sources of phenolic compounds.

For the first time, this study examined the effects of Vistula TC on human metabolome. The different doses of Vistula TC induced distinctive changes in the metabolome and showed the greatest shift following the HI dose. The LO dose provided an initial upregulation at 1 and 2 h (peaking at 2 h) but was reduced thereafter. The HI dose provided a slower upregulation of global metabolites; however, this upregulation was maintained for longer (peaking at 3 h) and still observed at 8 h. This provides a valuable dose–response relationship with a new Vistula variety of TC which can be implemented in supplementation strategies. When comparing the present results to previous work that examined the dose–response relationship [23], the low doses in both studies are in agreement, showing peaks in phenolic compounds/shifts in the metabolome at 1–2 h. Furthermore, both data suggest the shift is transient and tends to return towards baseline at 8 h following an acute dose. When polyphenols are ingested and metabolised in the liver, compounds absorbed into the blood undergo biotransformations such as sulphation, glucuronidation, methylation, and glycine-conjugation, so they can be excreted in urine or bile [4]. Acute doses of Montmorency TC increased the quantity of urine anthocyanin metabolites, C3G equivalents, and peonidin 3-glucoside equivalents [29]. In addition, some metabolites, including anthocyanins [41], are metabolised in the liver and recycled back into the small intestines through the enterohepatic cycle, which can result in a biphasic pharmacokinetic response [42,43], a response that was observed in the present study (Figure 2).

Metabolomic PLS-DA and heatmaps displayed the most discriminate metabolites responsible for the overall shifts in the metabolome across the three conditions (Figure 4 and Figure 5). Metabolites, such as cortisone, acetylcholine, dihydrothymine, as well as several amino acids, including L-cystine, leucine, were upregulated after the HI dose of TC but not placebo. Identifying the upregulation of cortisone, an anti-inflammatory metabolite [44], could help to explain the role TC has on positive health outcomes, such as reducing cardiovascular risk factors [10]. In addition, leucine (HI at 3 and 8 h) and betaine (HI at 8, LO at 1 and 5) are upregulated in HI dose at 8 h post ingestion. Leucine has been highlighted as particularly important for the stimulation of postprandial muscle protein synthesis through mechanistic target of rapamycin complex1 (mTORC1) signalling [45,46], which might help to support previous observations of improved recovery following strenuous exercise [18,19,20]. Furthermore, betaine is another metabolite which has been upregulated and been responsible for significant differences between the conditions (1 and 5 h LO dose, 8 h HI dose). When supplemented, betaine improved squat repetitions to fatigue [47] and improved bench press work capacity during cycles of high-volume training [48]. Identifying these metabolite shifts following acute consumption provides clues on the potential role of Vistula TC in exercise recovery.

Other metabolites such as uracil (3, 5, and 8 h HI dose) and dihydrothymine (1 h HI, 8 h LO) were increased at multiple timepoints in the LO and HI doses of Vistula TC but not in the placebo. Uracil participates in serval enzymatic reactions, and evidence indicates a relationship to various health conditions [49]. For instance, a review reported the close association between pyrimidine metabolism dysfunction and cancer progression [50]. Dihydrothymine is a marker of thymine synthesis and metabolism that is often used to evaluate the metabolic level of thymine [51]. It is thought to represent the presence of DNA damage [52] and play an important physiological role in skin cancer [53]. While there is no direct evidence linking TC intake to pyrimidine metabolism, the present study shows these compounds have increased following TC intake, and studies show TC intake might influence pathways related to inflammation, oxidative stress, and cellular signalling [54].

Acetylcholine was upregulated in both the LO (1 and 5 h) and HI dose (8 h), but not in the placebo. Acetylcholine serves as a neurotransmitter that triggers muscle contractions through synaptic transmissions at the neuromuscular junction [55] and highlights the potential mechanisms by which TC can attenuate the force reduction following strenuous exercise. Furthermore, acetylcholine plays a crucial role in cellular processes such as endocytosis, exocytosis, membrane fusion, and neurotransmitter uptake/release [56]. High-intensity exercise can reduce serum choline levels, leading to potential phosphatidylcholine mobilization, damaging membrane integrity, and causing muscle injury [57]. A decline in acetylcholine during intense exercise may result in reduced muscle contraction, accelerating muscle fatigue onset. Therefore, if TC consumption stimulates acetylcholine production, it could elucidate TC’s beneficial effects.

## 5. Conclusions

These results provide new insights on the dose- response of a powdered Vistula TC extract. Previous observations showing health and exercise benefits have attributed them to the uptake of phenolics; these data provide evidence that they are bioavailable and induce shifts in the metabolome. Although the changes could be attributed to TC uptake, the potential mechanisms of action for health benefits are not clear. These data could be used to inform future intervention studies where changes to physiological outcomes could be influenced by metabolomic shifts following acute supplementation. Additionally, the doses consumed had no side effects.

## Figures and Tables

**Figure 1 nutrients-16-01023-f001:**
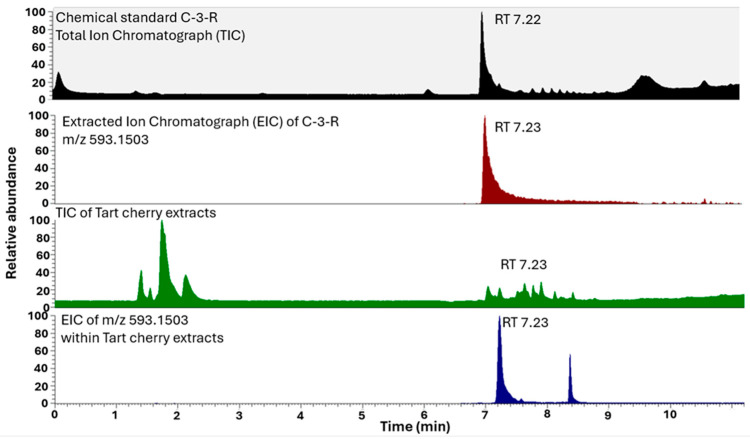
LCMS chromatographs of C3R (retention time 7.23 min) in the standard and the TC extract, extraction ion chromatogram of *m*/*z* 593.15033.

**Figure 2 nutrients-16-01023-f002:**
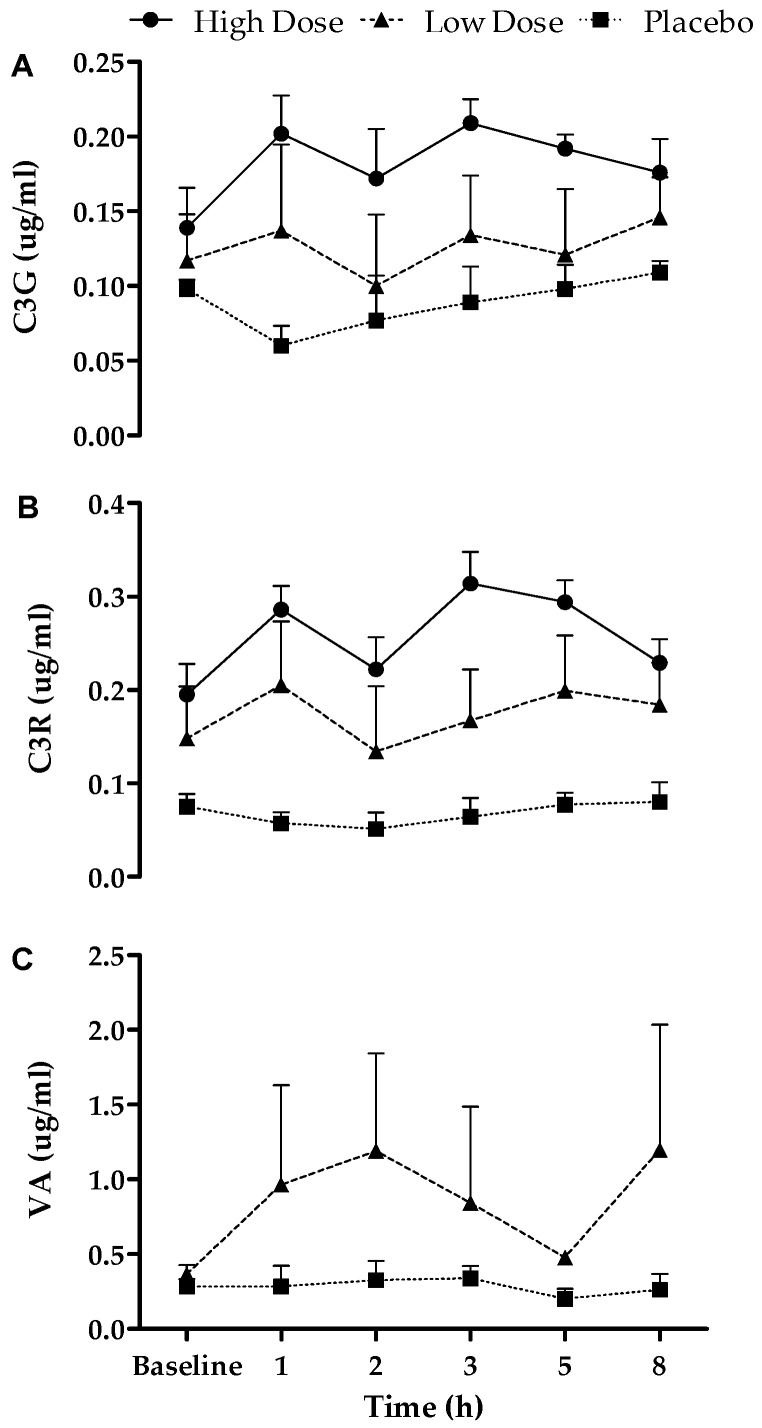
Bioavailability over 8 h in the HI, LO, and placebo dose for (**A**) cyanidine-3-*O*-glucoside (C3G); (**B**) cyanidine-3-*O*-rutinoside (C3R); (**C**) vanillic acid (VA), (*n* = 6); data presented as mean ± SD.

**Figure 3 nutrients-16-01023-f003:**
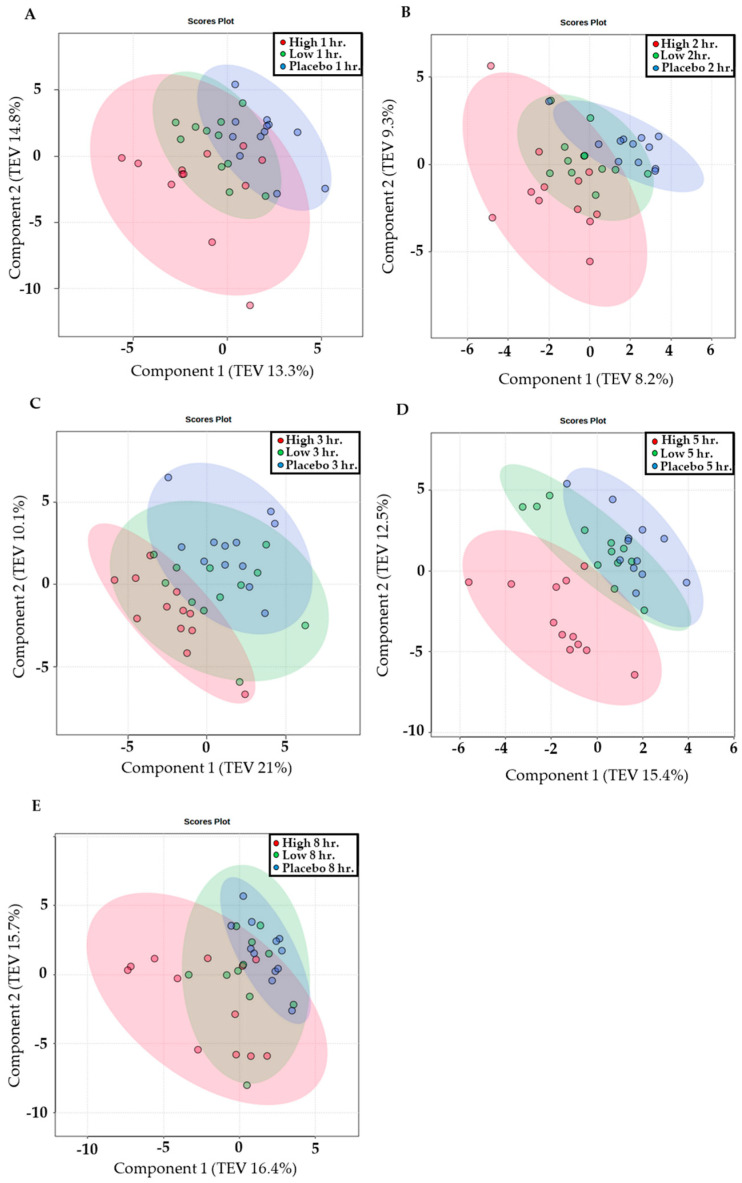
Supervised partial least square discriminate analysis (total explained variance (TEV %)) visualisation of the serum dose response (HI, LO, and placebo doses are represented in pink, green, and blue, respectively) at (**A**) 1 h, (**B**) 2 h, (**C**) 3 h, (**D**) 5 h, and (**E**) 8 h time points.

**Figure 4 nutrients-16-01023-f004:**
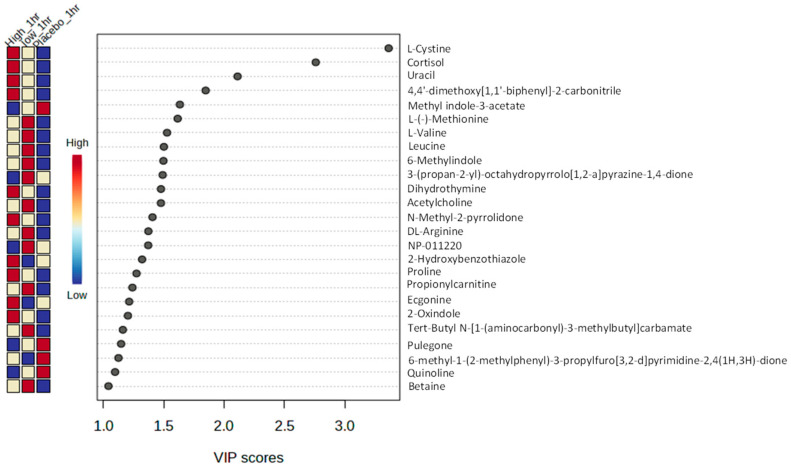
Exemplar plot of PLS-DA ranking of the dysregulate metabolites and VIP scores at 1 h (VIP score x > 1 is statistically significant).

**Figure 5 nutrients-16-01023-f005:**
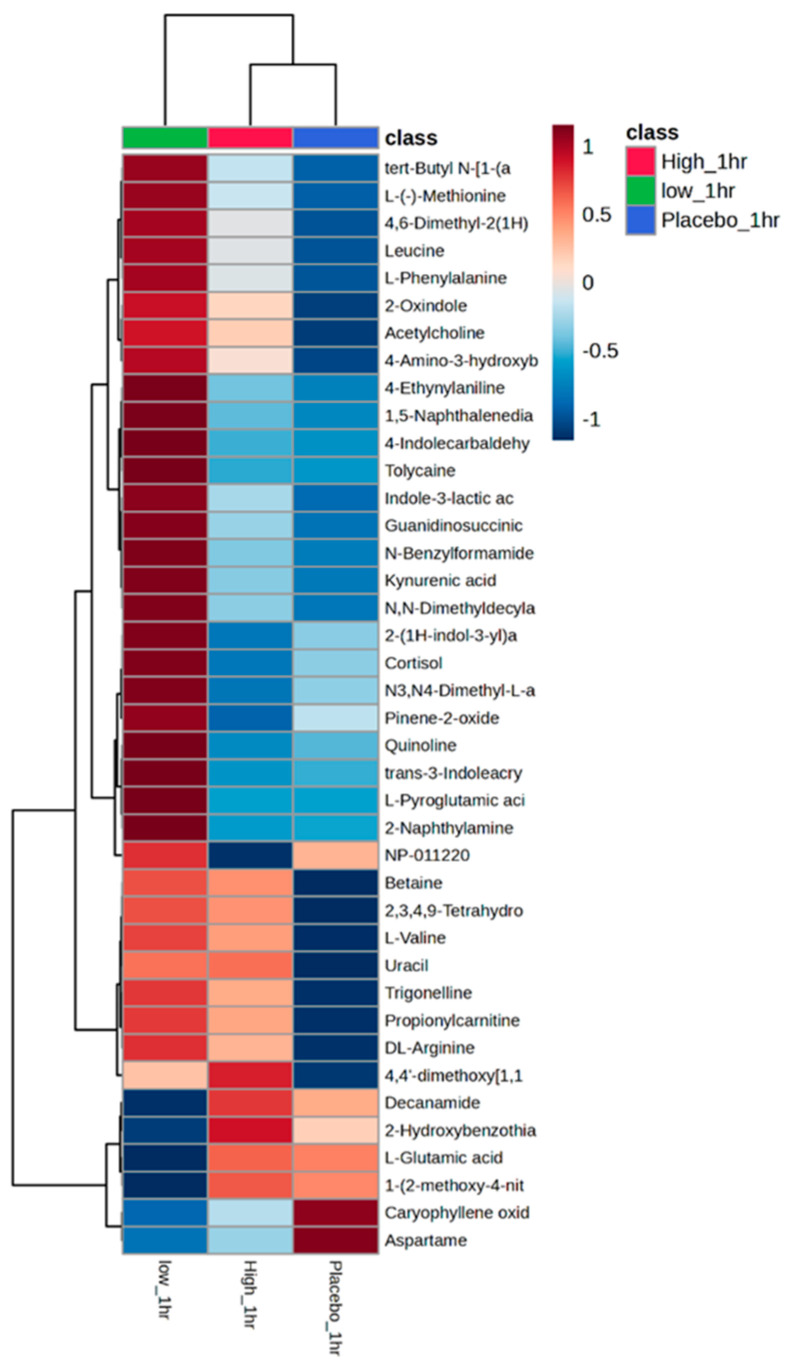
Exemplar of class average heatmap of the top 40 metabolites (ID using PLS-DA at 1 h).

**Table 1 nutrients-16-01023-t001:** The detection and relative abundance of compounds found in the extract.

	Chemical Formula	Molecular Weight	Monoisotopic Mass	M-H	Retention Time (min)	Relative Abundance
Protocatechuic Acid	C_7_H_6_O_4_	154.12	154.02661	153.0187	6.89	3.68 × 10^5^
Vanillic Acid	C_8_H_8_O_4_	168.14	168.04226	167.0347	8.12	3.75 × 10^3^
Chlorogenic Acid	C_16_H_18_O_9_	354.31	354.09058	353.0865	7.52	5.00 × 10^6^
Cyanidine-3-*O*-Rutinoside	C_27_H_31_O_15_	595.52	595.16630	593.1505	7.22	5.02 × 10^6^
Peonidin-3-*O*-Rutinoside	C_28_H_33_O_15_	609.6	609.18195	607.1655	7.47	5.68 × 10^5^
Cyanidine-3-*O*-Glucoside	C_21_H_21_O_11_	449.38	449.10839	447.0924	7.22	3.09 × 10^5^

## Data Availability

Data are kept on the University secure server in line with UK law relating to General Data Protection Regulations and the University’s Research Data Management Policy. Requests for data should be sent to the corresponding author.

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
