# Peer review of "Plasma-Induced Changes in the Metabolome Following Vistula Tart Cherry Consumption"

_nutrients, 2024, doi:10.3390/nu16071023_

Round 1

Reviewer 1 Report

Comments and Suggestions for Authors

nutrients-2896349
Title:
Plasma-Induced Changes in the Metabolome Following Vistula Tart Cherry
Consumption.
Summary:
The aim of the study is to examine the effects of acute Vistula TC consumption on changes in
the human metabolome. The authors suggest this could be used as basic information for future
intervention studies. The Vistula Tart Cherry is more abundantly used than the Montmorency
cultivar of TC.
The study is well written.
Major remarks:
No major remarks
Minor remarks:
1. Abstract: line 20: Change “overnight” to “overnight fast”.
2. Material & Method; Line 128. Unclear what is measured with the HPLC. And it is unclear if it
was qualitative or quantitative measurement.
3. Line 169: “Inclusion list”. List of metabolites that are measured should be available as
supplemental.
4. Results 3.2 and Figure 3. It is unclear in the legend with what dose (Low or High) a
significant effect is observed. By reading the result section it looks that sometimes the
effect is only in comparison with the High but not in the Low. Or in the Low and not in the
High. I suggest putting them both in the figures. Or make figures in which you compare
Placebo vs High and Placebo vs Low.
5. Figure 3. The Y-axis is “%”. % of what? Please add this information to the legend.

Author Response

Sincere thanks for your time and energy in providing a constructive and very timely review of the work. The comments were extremely helpful and as a result we have responded to all comments in a systematic way and where appropriate made changes to the manuscript. We amended the manuscript accordingly in line with suggestions. 

Reviewer 1

Comment 1. Abstract: line 20: Change “overnight” to “overnight fast”.

Response 1: Thank you for pointing this out. We agree with this comment and therefore we have edited line 20 to “overnight fast”.

Comment 2. Material & Method; Line 128. Unclear what is measured with the HPLC. And it is unclear if it was qualitative or quantitative measurement.

Response 2: We agree with your comment and have therefore clarified lines 140-141 by changing it to “on the plasma samples, for quantitative analysis of 3,4-dihydroxybenzoic acid, chlorogenic acid, VA, C3G, C3R, peonidin-3-O-rutinoside and melatonin,”.

Comment 3. Line 169: “Inclusion list”. List of metabolites that are measured should be available as supplemental.

Response 3: Thank you bringing this to our attention. The peak tables have been uploaded as part of the supplementary documents. We have subsequently amended lines 173-174 to include “(peak tables are available in the supplementary documents)”.

Comment 4. Results 3.2 and Figure 3. It is unclear in the legend with what dose (Low or High) a significant effect is observed. By reading the result section it looks that sometimes the effect is only in comparison with the High but not in the Low. Or in the Low and not in the High. I suggest putting them both in the figures. Or make figures in which you compare Placebo vs High and Placebo vs Low.

Response 4: We agree with your comment and have made the following amendments. In figure 2 we have now included the high and low dose as recommended for C3G and C3R. Due to sampling error in the high dose for VA, only LO and placebo are in the figure. This is made clear with the legend of figure 2. We have subsequently amended the text (lines 233-238 and 240), clarifying the comparisons. We pondered for some time about taking these data out altogether, but it seemed that it was more sensible to include the data, even though it is not complete.  In figure 3, we have clarified which colour represents which dose in the figure title (lines 250-253).

Comment 5. Figure 3. The Y-axis is “%”. % of what? Please add this information to the legend.

Response 5: Thank you for pointing this out, we have subsequently amended figure 3 axis titles and figure title (lines 252 – 255) to include this information.

Reviewer 2 Report

Comments and Suggestions for Authors

The paper "by E. Squires, I.H. Walshe, W. Cheung, S. Bowerbank, John R. Dean, Jacob Wood, M.P. McHugh, S. Plattner and G. Howatson entitled "Plasma-Induced Changes in the Metabolome Following Vistula Tart Cherry Consumption" is a valuable contribution to the field of nutrition and metabolism.

The authors conducted their study meticulously and commendably adhered to well-designed testing procedures that underpin the credibility of their findings. The detailed methodology presented in the paper not only demonstrates the thoroughness of the research but also provides valuable insights into the mechanisms underlying the effects of Vistula Tart Cherry (TC) consumption on the metabolome.

Although the sample size of 12 participants may seem relatively small, it must be acknowledged that this study represents the first exploration of the metabolic effects of TC consumption. The inclusion of comprehensive testing procedures and robust data analysis compensate for the limited sample size so that the results allow meaningful conclusions to be drawn.

The suggestion to include younger, physically active individuals in future studies is well-founded and offers a potential avenue to understand better how TC affects metabolism in different demographic groups.

In summary, this study, authored by E. Squires, I.H. Walshe, W. Cheung, S. Bowerbank, John R. Dean, Jacob Wood, M.P. McHugh, S. Plattner and G. Howatson, significantly advances our understanding of the metabolic effects of TC consumption. Their careful approach sets a high standard for future research in this area, and I anticipate further advances and insights from future studies inspired by this work.

Author Response

Sincere thanks for your time and energy in providing a constructive and very timely review of the work. The comments were extremely helpful and as a result we have responded to all comments in a systematic way and where appropriate made changes to the manuscript. We amended the manuscript accordingly in line with suggestions. 

Reviewer 2

Response 1: Sincere thanks for the complementary comments.  We are pleased the reviewer saw value in this work.

Reviewer 3 Report

Comments and Suggestions for Authors

The study aimed to investigate the impact of acute high- and low-dose tart cherry supplementation on the human metabolomic profile. In a randomized, double-blind, placebo-controlled crossover design involving 12 healthy participants, blood samples were taken at various intervals after consumption of tart cherry or placebo. Analysis revealed significant differences in metabolomic profiles between tart cherry supplementation and placebo, suggesting bioavailability of phenolics in plasma and potential modulation of the metabolome following acute tart cherry consumption, informing future intervention studies.

As a descriptive piece, I find this study to be highly informative, shedding light on the potential benefits of tart cherry consumption. However, I have three minor suggestions for improvement.

1. The quality of some figures could be enhanced, particularly in regard to the size and clarity of axis titles. For instance, Figure 3's x-axis and y-axis titles appear too small, and the font clarity of Figures S3, S5, S7, and S9 could be improved.

2. Consider incorporating volcano plots to present metabolomics data, providing readers with a broader understanding of metabolite changes in plasma.

3. Including raw metabolomic data in the supplement would increase the study's informativeness, enabling readers to delve deeper into the findings and potentially conduct further analysis.

Author Response

Sincere thanks for your time and energy in providing a constructive and very timely review of the work. The comments were extremely helpful and as a result we have responded to all comments in a systematic way and where appropriate made changes to the manuscript. We amended the manuscript accordingly in line with suggestions. 

Reviewer 3:

Comment 1. The quality of some figures could be enhanced, particularly in regard to the size and clarity of axis titles. For instance, Figure 3's x-axis and y-axis titles appear too small, and the font clarity of Figures S3, S5, S7, and S9 could be improved.

Response 1: Thank you for pointing this out. In line with comments for other reviewers, we collectively agree with this comment and therefore we have improved the quality and enlarged the figures in the supplementary documents (namely, S3, S5, S7 and S9). As for figure 3, we have amended the axis titles (lines 252-255).

Comment 2. Consider incorporating volcano plots to present metabolomics data, providing readers with a broader understanding of metabolite changes in plasma.

Response 2: Thank you for this comment. We have reviewed the suggestion to in corporate volcano plots, however we feel the PLS-DA VIP and the relative abundance heatmaps can show this relationship more effectively. While volcano plots can be effectively applied to a 2 class comparison, here with a 3 class comparison, it would require multiple volcano plots to be generated for comparison (i.e. placebo vs low and placebo vs high, respectively), which we think would detract from the message of the paper.

Comment 3. Including raw metabolomic data in the supplement would increase the study's informativeness, enabling readers to delve deeper into the findings and potentially conduct further analysis.

Response 3: Thank you for your comment. We agree and in line with a previous reviewers comments  we have included the peak tables as part of the supplementary documents and amended lines 173-174 to include “(peak tables are available in the supplementary documents)”.  We think this was a great idea to share this with the wider community.